# IMM-H004 Protects against Cerebral Ischemia Injury and Cardiopulmonary Complications via CKLF1 Mediated Inflammation Pathway in Adult and Aged Rats

**DOI:** 10.3390/ijms20071661

**Published:** 2019-04-03

**Authors:** Qidi Ai, Chen Chen, Shifeng Chu, Yun Luo, Zhao Zhang, Shuai Zhang, Pengfei Yang, Yan Gao, Xiaoling Zhang, Naihong Chen

**Affiliations:** 1Hunan Engineering Technology Center of Standardization and Function of Chinese Herbal Decoction Pieces & College of Pharmacy, Hunan University of Chinese Medicine, Changsha 410208, China; 13357329630@163.com; 2State Key Laboratory of Bioactive Substances and Functions of Natural Medicines, Institute of Materia Medica & Neuroscience Center, Chinese Academy of Medical Sciences and Peking Union Medical College, Beijing 100050, China; chenchen@imm.ac.cn (C.C.); chushifeng@imm.ac.cn (S.C.); zhangzhao@imm.ac.cn (Z.Z.); zhangshuai2586@163.com (S.Z.); yangclatter@163.com (P.Y.); gaoyantalentback@126.com (Y.G.); zxl123456mm@163.com (X.Z.); 3Institute of Medicinal Plant Development, Peking Union Medical College and Chinese Academy of Medical Sciences, Beijing 100193, China; ly20040423@126.com

**Keywords:** IMM-H004, cerebral ischemia, complication, CKLF1, inflammation, aged rats

## Abstract

(1) Background: Chemokine-like factor 1 (CKLF1) is a chemokine with potential to be a target for stroke therapy. Compound IMM-H004 is a novel coumarin derivative screened from a CKLF1/C-C chemokine receptor type 4 (CCR4) system and has been reported to improve cerebral ischemia/reperfusion injury. This study aims to investigate the protective effects of IMM-H004 on cerebral ischemia injury and its infectious cardiopulmonary complications in adult and aged rats from the CKLF1 perspective. (2) Methods: The effects of IMM-H004 on the protection was determined by 2,3,5-triphenyltetrazolium chloride (TTC) staining, behavior tests, magnetic resonance imaging (MRI) scans, enzyme-linked immunosorbent assay (ELISA), Nissl staining, histo-pathological examination, and cardiopulmonary function detection. Immunohistological staining, immunofluorescence staining, quantitative real-time PCR (qPCR), and western blotting were used to elucidate the underlying mechanisms. (3) Results: IMM-H004 protects against cerebral ischemia induced brain injury and its cardiopulmonary complications, inhibiting injury, and inflammation through CKLF1-dependent anti-inflammation pathway in adult and aged rats. IMM-H004 downregulates the amount of CKLF1, suppressing the followed inflammatory response, and further protects the damaged organs from ischemic injury. (4) Conclusions: The present study suggested that the protective mechanism of IMM-H004 is dependent on CKLF1, which will lead to excessive inflammatory response in cerebral ischemia. IMM-H004 could also be a therapeutic agent in therapy for ischemic stroke and cardiopulmonary complications in the aged population.

## 1. Introduction

Acute ischemic stroke is a devastating and debilitating disease, leading to high morbidity and mortality worldwide [1]. Aging is the biggest risk factor of stroke [2], with incidence doubled every decade after 55 years old [3]. The International Stroke Trial, including more than 17,000 stroke patients, revealed 84.1% of enrolled subjects were over 60 years old [4]. The average age of first stroke onset was at about 70 years old in very large reports, highlighting the large aged population of stroke patients [5,6].

Thrombolytic therapy is commonly used in patients with acute ischemic stroke, and recombinant tissue plasminogen activator (tPA) is the only Food and Drug Administration (FDA)-approved drug in clinical application. However, it is widely known that thrombolytic therapy only beneficial within 4.5 h after ischemic onset [7], but also with high incidences of side effects—such as causing hemorrhagic transformation or augmenting the inflammatory response [8,9]. In present clinical practice, many patients cannot be administered tPA for exceeding the narrow time window, resulting in permanent cerebral ischemic injury. Therefore, it is necessary and urgent to develop safer and more effective therapeutic agents for ischemic stroke in the aged population.

Post-stroke complications are major contributors to the high mortality rate of ischemic stroke [10,11]. Long-term clinical observations have indicated that the complications are concentrating on heart and lung organ dysfunctions, including myocardial infarction (MI), pulmonary embolism (PE), and pneumonia, which are also the most common causes for ischemic stroke death [10,11,12,13,14,15]. Cardiovascular disease is the main cause of death from the second to the fifth year after stroke [16,17]. The immune system is known to participate in ischemic brain injury, and the most common infection after stroke is pneumonia, which has a three-fold increase in the risk of death [18,19,20,21,22,23]. In addition, the aged population is more prone to post-stroke complications due to the weaker immune system compared with young people [1]. However, there are no effective drugs available for both cerebral injury and complications in ischemic stroke therapy to date. Research of ischemic stroke treatment mainly focuses on protection against brain injury but few on post-stroke complications, thus, developing novel drugs with multiple function for cerebral damage and cardiopulmonary complications is extremely important.

Chemokine-like factor 1 (CKLF1) is a chemokine with C-C chemokine receptor type 4 (CCR4) as a receptor. Current research mainly focuses on the role of CKLF1 in periphery infectious diseases [24,25,26,27]. Our previous studies have found a significant increase of CKLF1 expression in damaged brain as early as 12 h after middle cerebral artery occlusion (tMCAO), which peaks at day 2 after tMCAO in cerebral ischemic rats [28]. Intracerebroventricular-targeted injection of C19, an antagonist peptide of CKLF1, can improve cerebral ischemia injury in rats [29]. In addition, administration of anti-CKLF1 antibody also shows beneficial effects on smaller infarctions and better neurological behavior in rats [30].

Although the mechanism of acute ischemic stroke induced pneumonia and cardiovascular complications is still unclear, studies have shown that it may have a close association with inflammation [31,32]. As a chemokine, CKLF1 may play an important role in immune response after stroke. Thus, whether CKLF1 involves in the stroke induced cardiopulmonary complications is of considerable interest. IMM-H004 is a novel 3-piperazinylcoumarin small molecule compound screened from a CKLF1/CCR4 system by calcium transient technology and modified [27,33]. Our previous work has demonstrated protective effects of IMM-H004 on cerebral ischemia. It can protect against global cerebral ischemia by suppressing apoptosis and maintaining the integrity of synaptic structure in adult rats [34], improving BBB function via inhibiting the release of inflammatory factors in mice [35,36]. IMM-H004 can also significantly ameliorate cerebral ischemia/reperfusion-caused injury and subsequent inflammation in spontaneously hypertensive rats [37]. Additionally, IMM-H004 could reduce tPA’s side effects by improving energy metabolism when combined with tPA [38]. However, it is undetermined whether IMM-H004 has curative effect on aged rats with permanent cerebral ischemia injury and especially the accompanied complications. There are also no studies investigating the change of CKLF1 when administered with IMM-H004. Therefore, our aim in the present study is to determine the therapeutic effects of IMM-H004 on permanent focal cerebral ischemia and cardiopulmonary complications in adult and aged rats, elucidating the underlying mechanisms from the CKLF1 pathway.

## 2. Results

### 2.1. Effects of IMM-H004 on Permanent Focal Cerebral Ischemia-Induced Brain Injury in Adult Rats

The protective effect of IMM-H004 on permanent focal cerebral ischemia-induced brain injury is determined. Firstly, the therapeutic time window of IMM-H004 was investigated. The infarct size of a representative rat assessed by the TTC assay is shown in Figure 1C. Compared with the pMCAO-operated group, rats administered with IMM-H004 (10 mg/kg) showed smaller brain infarct size and ameliorated neurological deficits pronouncedly at 3 h and 6 h after ischemia, but IMM-H004 administration showed no protective effect at 9 h and 12 h after ischemia. These results showed that IMM-H004 (10 mg/kg) treatment could decrease permanent focal cerebral ischemia-induced brain injury, and its therapeutic time window for ischemia is 0 to 6 h (Figure 1C,D). In order to exclude the impact of IMM-004 on the sham group, we set up the sham combined with IMM-H004 group to compare with the sham group. The results showed that there were no significant differences between the sham group and sham combined with IMM-H004 group in parameters we checked (infarction by TTC staining, neurobehavioral deficit tests, and inflammatory cytokines by ELISA analysis) (Appendix A).

Secondly, we determined the therapeutic dosage window of IMM-H004. Representative TTC staining for brains of rats administrated with different dosages of IMM-H004 were shown in Figure 2B. pMCAO model was built on aged rats and IMM-H004 (10 mg/kg) was administrated at 6 h after ischemia, and rats were sacrificed 3 h after drug administration in all the following experiments. IMM-H004 significantly reduced the brain infarction and neurological dysfunction in rats with dosages of 5, 10, and 20 mg/kg compared with the rats received saline (Figure 2A). These results showed that the therapy dosage window of IMM-H004 is 5 to 20 mg/kg (Figure 2B,C).

To investigate whether IMM-H004 has protective effects beyond 6 h after ischemia, we performed continuous administration of IMM-H004 1 time daily for 3 days. The first administration was at 6 h after pMCAO, and we defined this time point as 0 h, and the second and third administrations were at 24 and 48 h. The survival rate and behavioral scores were counted at 0, 24, 48, and 72 h (Figure 3A). Rats were randomly assigned to four groups: 1, pMCAO insult; 2, pMCAO combined with IMM-H004 (10 mg/kg); 3, pMCAO combined with edaravone (10 mg/kg); 4, pMCAO combined with urokinase (10,000 UI/kg). All of the rats were sacrificed at 72 h. As shown in Figure 3B–D, the 72 h survival rate of IMM-H004 treatment was better than urokinase treatment and the same as edaravone treatment. IMM-H004 and urokinase treatment can significantly improve the neurological deficits compared with that of the model group. TTC staining showed no significant difference among each group at 72 h after ischemia. These results demonstrated that continuous administration of IMM-H004 (10 mg/kg) showed improved survival compared to edaravone and urokinase after permanent focal cerebral ischemia in rats, however, the infarct size remains unchanged (Figure 3B–D).

Moreover, since clinical application for stroke therapy are frequently multiple drug administration daily, effects of IMM-H004 by repeated administration in one day was also studied. Drugs were administered at 3, 6, and 12 h after ischemia (0 h). MRI and behavior tests were performed at 0, 3, 6, 12, and 24 h after ischemia (Figure 4A). The results showed that IMM-H004 (10 mg/kg) multiple dosing administration also pronouncedly reduced the brain infarct size and attenuated neurological deficits (Figure 4B,C).

### 2.2. Effects of IMM-H004 on Permanent Focal Cerebral Ischemia Induced Brain Injury in Aged Rats

To investigate whether IMM-H004 has protective effects on aged rats, pMCAO model was built on aged rats and IMM-H004 (10 mg/kg) was administrated at 6 h after ischemia, and rats were sacrificed 3 h after drug administration in all the following experiments. A total of 45 rats were randomly assigned to three groups: 1, sham group; 2, pMCAO insult group; 3, pMCAO combined with IMM-H004 (10 mg/kg) treatment group. MRI scanning images showed that IMM-H004 treatment pronouncedly reduced the brain infarct size and edema volume (Figure 5A).

Moreover, after permanent focal cerebral ischemia, the number of normal CA1 hippocampus neurons, cortex neurons, and striatum neurons of the pMCAO treatment group sharply decreased, whereas the relict neurons showed karyopyknosis, anachromasis, nucleoli disappeared and vacuolization as assayed by Nissl staining (Figure 5B). IMM-H004 significantly protected the number of neurons in these brain regions from the stroke-induced reduction (Figure 5B).

### 2.3. IMM-H004 Inhibits Brain Inflammation after Permanent Focal Cerebral Ischemia in Aged Rats

To elucidate the anti-inflammation mechanism of IMM-H004 in permanent focal cerebral ischemia-induced brain injury of aged rats, the expression of CKLF1 in hippocampus, cortex, and striatum of brain were assayed by immunohistological staining. The expression of CKLF1 was low in the sham group, however, CKLF1 was increased significantly in pMCAO treatment group, and treatment with IMM-H004 (10 mg/kg) significantly decreased the expression of CKLF1 (Figure 6). The expression of CCR4 was detected by immunofluorescence staining. The results showed that the expression levels of CCR4 in the hippocampus, cortex, and striatum of aged rat have no significant change among all the groups (Figure 7A).

Inflammatory cytokines in the hippocampus, cortex, and striatum of aged rats were assayed by ELISA to determine the anti-inflammation effects of IMM-H004. The expression of IL-1β and TNF-α in hippocampus, cortex, and striatum of aged rats were all markedly increased in response to pMCAO operation, while these cytokines were significantly reduced by treatment with IMM-H004 (10 mg/kg). Similar effects were shown on adult rats (Figure 7B).

As assayed by qPCR, the mRNA expression levels of CKLF1, IL-1β, and TNF-α in the hippocampus, cortex, and striatum of aged rats were markedly increased by pMCAO treatment and that the expression levels were decreased in the IMM-H004 (10 mg/kg) treatment group, and equally, the expression levels of CCR4 showed no differences among all of the groups (Figure 8A). The protein expression of CKLF1, CCR4, p-NF-κB, NF-κB, IL-1β, and TNF-α in the hippocampus, cortex, and striatum of brain were detected by western blotting. IMM-H004 (10 mg/kg) treatment can decrease the expression levels of CKLF1, p-NF-κB, IL-1β, and TNF-α, which were increased by pMCAO treatment, whereas the expression levels of CCR4 showed no differences among all of the groups (Figure 8B).

### 2.4. Effects of IMM-H004 on Cardiopulmonary Complications in Aged Rats

An overall view of the distribution of myocardial and lung damage of aged rats by HE staining was shown in Figure 9A. No obvious abnormalities were observed in the sham group, whereas myocardial damages were found in the pMCAO operated group characterized by eosinophilic changes in intensity, cytoplasmic vacuolization, and contraction band anomaly. IMM-H004 (10 mg/kg) treatment ameliorated the myocardial damages significantly. Similarly, no obvious abnormalities were observed in the lung of rats with sham operation, whereas remarkable lung damages were found in rats suffered with pMCAO operation, which characterized by inflammatory cell infiltration, alveolar wall thickening, pulmonary interstitial edema, and hemorrhage. Treatment with the IMM-H004 (10 mg/kg) could prevent lung damage markedly. LDH leakage in heart was also determined to evaluate the cardiac damage. As shown in Figure 9B, pMCAO operation induced a huge release of LDH, and IMM-H004 (10 mg/kg) treatment could significantly inhibit the release of LDH.

### 2.5. IMM-H004 Suppresses Inflammation in Heart and Lung Post-Stroke of Aged Rats

To determine the changes of inflammatory response in heart and lung, the inflammatory cytokines were detected. We found that secreted IL-1β and TNF-α levels in heart and lung were all markedly increased after pMCAO operation, while these inflammatory cytokines reduced significantly by IMM-H004 administration (Figure 10B).

Then, the expression of CKLF1 was also determined in heart and lung by immunohistological staining. The results showed that expression of CKLF1 in heart and lung of the aged rats were both significantly increased after pMCAO operation. IMM-H004 treatment significantly decreased the expression of CKLF1 in heart and lung (Figure 9C). CCR4 expression of heart and lung (Figure 10A) showed neither significant changes after pMCAO operation, nor changed by treatment with IMM-H004 as assayed by immunofluorescence staining. The cardiopulmonary function detection including left ventricular systolic pressure (LVSP), left ventricular developed pressure (LVDP), the maximal ventricular pressure rise ratio during systolic period (+dp/dtmax), the maximal ventricular pressure decrease ratio during diastolic period (-dp/dtmax), minute ventilation (VE), and forced expiratory volume in the first second (FEV1) were determined in adult and aged rats. This was done at 9 h post-stroke (3 h post-drug). LVSP, +dp/dtmax, -dp/dtmax, VE, and FEV1 were markedly increased by pMCAO treatment and decreased by IMM-H004 while LVDP was opposite (Figure 11). As for the post-stroke infection, inflammation has been detected in the organ (Figure 10B). Because of the early complications of our detection, the observation time is 9 h after ischemia, and the infection may not be very serious. We will observe the relevant indicators in the further research. The mRNA and protein expression levels of CKLF1, CCR4, p-NF-κB, NF-κB, IL-1β, and TNF-α in heart and lung were determined. CKLF1, IL-1β, p-NF-κB, NF-κB, and TNF-α markedly increased by pMCAO treatment, and decreased by IMM-H004. The expression levels of CCR4 showed no differences among all of the groups (Figure 12A,B).

### 2.6. IMM-H004 Exerts Protective Effects in Ischemic Brain through CKLF1 Mediated Inflammatory Pathway

To investigate whether CKLF1 is necessary for IMM-H004 to exert its potency and elucidate the protective mechanism of IMM-H004, CKLF1^−/−^ rats were used. As shown in Figure 12, the pMCAO insult induced brain infarction and neurological deficits in CKLF1 deficient rats, but IMM-H004 showed no protective effects against brain damage on CKLF1^−/−^ rats (Figure 13A,B). These results demonstrated that IMM-H004 had no neuroprotective effect when CKLF1 is loss, suggesting the efficacy of IMM-H004 for brain protection is directly through targeting the CKLF1 (Figure 13A,B).

p-NF-κB, NF-κB, IL-1β, and TNF-α in brain of CKLF1^−/−^ rats were markedly increased by pMCAO insult, but IMM-H004 induced reduction in these increased expression levels was abolished in CKLF1^−/−^ rats (Figure 14A,B). The mRNA and protein expression levels of CCR4 showed no differences among all of the groups (Figure 14A,B). These results demonstrated that the protective effects of IMM-H004 against ischemic stroke-induced brain injury and inflammation is through the CKLF1 pathway involved with NF-κB.

### 2.7. IMM-H004 Protects against Ischemic Stroke Induced Cardiopulmonary Complications via CKLF1

In CKLF1^−/−^ rats, the expression levels of p-NF-κB, NF-κB, IL-1β, and TNF-α in heart and lung were all markedly increased by pMCAO treatment, but showed no decrease by IMM-H004 administration. Equally, the expression levels of CCR4 in mRNA and protein levels did not disturbed by pMCAO or treatment with IMM-H004 (Figure 15A,B). As verified by CKLF1^−/−^ rats, the protective effects of IMM-H004 against permanent focal cerebral ischemia-induced cardiopulmonary complications is via the CKLF1 inflammation pathway involved with NF-κB.

## 3. Discussion

Stroke remains a leading cause of death worldwide despite early revascularization therapies and progress in drug development. The main cause of death is stroke induced complications, manifested by heart and lung organ dysfunction. At present, no effective drugs are available for treatment of stroke and cardiopulmonary complications simultaneously. In present study, we found that IMM-H004 exerts protective effects on brain infarction and damage in heart and lung in a pMCAO rats model (Figure 1, Figure 2, Figure 3, Figure 4, Figure 5, Figure 6, Figure 7, Figure 8, Figure 9, Figure 10, Figure 11 and Figure 12). Moreover, the therapeutic time window of IMM-H004 on permanent focal cerebral ischemia was determined to be 0–6 h which is wider than that of tPA. In order to extend the therapeutic time window, multiple drug administration daily was used. Figure 4 showed that IMM-H004 (10 mg/kg) multiple dosing administration also pronouncedly reduced the brain infarct size and attenuated neurological deficits. Because the protection mechanism of IMM-H004 is not thrombolytic therapy but anti-inflammation, so multiple dosing administration still have protective effect in long time beyond 6 h. In the further research we will investigate the effects of multiple administrations for a longer period of time beyond 24 h. Urokinase and edaravone are common clinical cerebral ischemia treatment drugs at present. Urokinase is a thrombolytic drug and edaravone is a strong antioxidant drug. Urokinase was chosen as the positive drug in this study for its therapeutic time window is 0–6 h same as the IMM-H004 [39,40]. We also chose edaravone as the positive drug which is widely used in ischemic stroke research [41,42,43,44].

The elderly are more prone to get ischemic stroke and vulnerable for stroke complications, we enrolled the aged rats in our study to investigate the efficacy of IMM-H004. Fortunately, IMM-H004 showed protective effects on both brain injury and cardiopulmonary complications in aged rats. These results suggested a potential for IMM-H004 to treat ischemic stroke in both adult and elderly. Additionally, IMM-H004 could not only protect the brain against damage, but also reduce the severity of cardiopulmonary complications. In clinical patients, stroke associated myocardial infarction and pneumonia mainly develop first weeks [14,23]. This work firstly reported that 9 h of ischemia could induce cardiopulmonary complications in aged rats, which is shorter than in human.

There are many theories for mechanisms of cerebral ischemia onset and progress, and inflammation is of particular interest at present [45]. IMM-H004 treatment could inhibit inflammatory response in brain of rats, as evidenced by the decrease of IL-1β and TNF-α in brain. Studies have shown that the cardiopulmonary complications after ischemic stroke is closely associated with inflammation [31,32]. We found that IMM-H004 inhibits the inflammation in heart and lung of ischemic rats. These results suggested that IMM-H004 exerts protective effects on brain injury and cardiopulmonary complications is partly through suppressing the inflammatory response after stroke.

IMM-H004 is a compound screened from a system of C27 (CKLF1 agonist peptide) and CCR4, and our previous studies have shown that CKLF1 may play an important and detrimental role in the early phase of a stroke [28,29,30,46,47]. Whether IMM-H004 is exert its protective effects through the CKLF1 is still unknown. In addition, whether CKLF1 is involved in the occurrence of cardiopulmonary complications is of our interest. In the present study, we found CKLF1 increased in ischemic brain and damaged heart and lung significantly after stroke, and IMM-H004 could decrease the expression of CKLF1 in these parts. As a receptor of CKLF1, CCR4 showed no obvious change among rats in sham, ischemia or IMM-H004 treatment groups. By the application of CKLF1^−/−^ rats, we confirmed that IMM-H004 shows protective effects is dependent on CKLF1. When CKLF1 is loss, its protective effects will also lose. Moreover, increased CKLF1 in heart and lung after ischemic stroke also suggested us CKLF1 may involve in the cardiopulmonary complications.

Immunosuppression plays a major role in cerebral ischemia injury induced complication, and is now more universally acknowledged to be the main explanation for susceptibility to infection after stroke [48,49]. The suppression of systemic immunity by the nervous system is thought to protect the brain from further inflammatory insult, yet this comes at the cost of increased susceptibility to infection after stroke. With regard to internal causes, stroke-mediated immunodeficiency increases the risk of infectious episodes, predominantly chest infection, within the first few days after stroke [50,51]. Immunodeficiency develops as early as 12 h after ischemic stroke and may persist for several weeks [52,53]; therefore, it plays a major role in causing pneumonia in post-stroke patients. Infection could trigger downstream inflammatory signaling cascades. A recent study indicates that experimental stroke models have indirectly revealed the pathologic effects of the ischemic brain on the heart. It suggests that inflammation factors are secreted by injured neuronal cells in the early stage of ischemic insult and subsequently reach the heart through the bloodstream in the later stages of stroke [32]. Inflammation is a key progress at the initiation of ischemia and post-stroke, involving a variety of immune cells and factors. Chemokines are a family of chemotactic cytokines that originally identified as factors attracting circulating leukocytes toward the site of inflammation or injury [54,55]. Chemokines generally play roles through interaction with their cognate receptors expressing on leukocytes and other cell type. As a chemokine, CKLF1 may involve in inflammatory response after stroke. In present study, we found that the increased CKLF1 after ischemic stroke is accompanied by excessive inflammation, both in the brain and periphery organs. IMM-H004 could both decrease the expression of CKLF1 and inflammatory cytokines. However, in CKLF1^−/−^ rats, IMM-H004 could not decrease the inflammatory cytokines anymore, which means the CKLF1 could mediate the followed inflammation partly. Interestingly, we found the CKLF1^−/−^ rats showed smaller infarction and lighter neurological deficient compared with the CKLF1^−/−^ rats. This finding indicated that CKLF1 plays a detrimental role after ischemia at least in our detected time points. NF-κB is considered the central transcription factor of inflammatory mediators [56,57]. Numerous studies have shown the cerebral ischemia will result in NF-κB activation [58,59]. In our present study, the same phenomenon was observed. NF-κB activated after ischemia, and IMM-H004 could inhibit the activation. In CKLF^−/−^ rats, although significant NF-κB activation is observed after cerebral ischemia, IMM-H004 showed no effect on NF-κB activation, which implies the CKLF1 mediated inflammation may be via the NF-κB pathway.

This study confirmed the protective effects of IMM-H004 on adult and aged rats, targeting brain injury and cardiopulmonary complications simultaneously. Moreover, IMM-H004 exerts its protective effects via inhibiting the CKLF1 mediated inflammation pathway, involving with NF-κB. Despite the findings which have significance for the novel drug development in present study, it still has some limitations. Firstly, despite in clinical investigations or basic animal studies, females showed significant difference in incidence, mortality, and morbidity of stroke compared with males [60,61]. In the present study, we used only male subjects to minimize confounding variables, such as hormonal cycling and estropause for this first series of experiments. We are very aware of potential sex differences in stroke outcome [62] and in immune responses [63,64]. Our future research will enroll sex as a biological variable to clarify CKLF1 functions in ischemia stroke. Secondly, the upstream transcription factors of CKLF1 need to be determined. Thirdly, despite NF-κB, there are other pathways involved in the downstream influence inflammatory response. These problems need to be resolved in further studies to strengthen the theoretical foundation of IMM-H004 and promote its development.

## 4. Materials and Methods

### 4.1. Experimental Animals

All of the experimental protocols and animal care were performed according to the National Research Council’s Guide for The Care and Use of Laboratory Animals. The project identification coed is 000353, which was approved by the Institutional Animal Care and Use Committee of the Peking Union Medical College and Chinese Academy of Medical Sciences in 10 September 2017. Male adult SD rats (6–8 weeks, 250–280 g) and male aged SD rats (26 months old, 500–600 g) were obtained from VITAL RIVER Laboratories, Beijing, China. CKLF1^−/−^ rats (250–280 g) were obtained from Key Laboratory of Human Disease Comparative Medicine, NHFPC, Institute of Laboratory Animal Science, Peking Union Medicine College, Chinese Academy of Medical Sciences, Beijing, China. All of the rats were housed under standard conditions. For the sample size calculation, preliminary experiment indicated that there was at least 10% difference in infarct volume between groups with standard deviation associated with these measurements to be 0.06. Therefore, we required a minimum of six animals per group to detect such a difference at 95% confidence (α = 0.05) and 0.8 power. The study was not pre-registered prior to examination of the data or observing the outcomes.

### 4.2. Materials

The compound IMM-H004 (molecular formula: C_16_H_20_O_4_N_2_; molecular weight: 304; purity > 99%) (Figure 1A) was provided by the Department of Chemosynthesis, Institute of Materia Medica, Chinese Academy of Medical Sciences and Peking Union Medical College (Beijing, China); all of the IMM-H004 in this paper is IMM-H004 citrate. Positive drugs: edaravone (C_10_H_10_N_2_O) was purchased from Simcere (Nanjing, China), and urokinase (C_21_H_25_BrN_2_O_3_) was purchased from the National Institutes for Food and Drug Control (Beijing, China). These compounds were dissolved in physiological saline for in vivo experiments.

### 4.3. Rat pMCAO Model and Drug Administration

Permanent focal cerebral ischemia was induced by permanent middle cerebral artery occlusion (pMCAO) for 0, 3, 6, 9, 12, 24, or 72 h, as described previously (Figure 1B) [38]. Anesthesia was maintained with 2% isoflurane (RWD, Shenzhen, China) in nitrous oxide/oxygen (70:30). Briefly, the internal carotid artery (ICA) and pterygopalatine artery of the ICA were carefully isolated. The left external carotid artery (ECA) was ligated far away from the bifurcation. To occlude the ICA, a 0.38 mm nylon filament (Cinontech Co., Beijing, China) was inserted from the left common carotid artery (CCA) to the ICA through a small incision and then advanced approximately 18 mm into the Circle of Willis until a faint resistance was felt. Successful pMCAO (at least 70% reduction of MCA flow compared to baseline) was confirmed by laser Doppler flowmetry (PeriFlux System 5000; Perimed, Stockholm, Sweden). Sham-operated rats underwent the same surgical procedure except for suture insertion. The rats were returned to their heated cages with free access to water and food after operation. After specific ischemic time, the available animals were further selected according to the neurological score. The rats were returned to their heated cages with free access to water and food after operation. After specific ischemic time, the available animals were further selected according to the neurological score. The neurological deficits were assessed by Zea longa test in a blinded manner as previously described [65]. 0: no neurological deficit; 1: failure to extend left forepaw fully; 2: circling to the left; 3: falling to the left; 4: failure to walk spontaneously or no consciousness; 5: death. Rats with neurological scores of 1–4 were enrolled for further treatment. All drugs were administered by intravenous injection at according time points of ischemia. There was no sample size difference between the beginning and end of the experiments.

### 4.4. TTC Staining and Behavior Assessment

TTC (Sigma-Aldrich, St. Louis, MO, USA) staining was used for determining the infarct areas as previously described [66]. Brains were removed instantly at 3 h post drug administration, and six consecutive coronal slices (2 mm) were prepared and stained with 1.5% TTC in phosphate-buffered saline (PBS) for 15 min and then fixed in 4% formaldehyde overnight. The infarct area was analyzed using IMAGE-PRO PLUS 6.0 (Media Cybernetics, Silver Springs, MD, USA) software: infarct area (%) = total infarct area/total section area ×100%; edema ratio (%) = left brain area/right brain area ×100% [67]. The neurological deficits of rats were assessed with Zea Longa test, Grid test, hanging test, and screen test in a blinded manner. 

The purpose of the grid test is to evaluate the motor ability of rats. Animals placed on an elevated (1 m) grid surface platform with square openings of 9 × 9 cm were encouraged to traverse the grid surface for one minute. The foot faults where animals inaccurately placed a limb through one of the openings in the grid were counted. The number of foot faults made per meter in one minute was calculated. The purpose of the hanging test is to check grasping ability. A stainless steel bar (50 cm in length) resting on two vertical supports and elevated 60 cm above a flat surface was used in this test. Rats were placed on the bar midway between the supports and were observed for 120 s. The amount of time spent hanging was recorded. 1: ≤10 s; 2: 11 s–30 s; 3: 31 s–2 min; 4: 3 min–5 min; 5: >5 min. Screen training: The screen is 50 × 40 cm mesh belt. The mesh is 1 × 1 cm. The left and right sides of the stencil are made of a 25-cm high wooden board. The height of the screen is 80 cm from the ground. The bottom is covered with 12-cm thick sponge. We first placed the screen horizontally, then placed the mouse on it. We then slowly raise one end, and turn the screen into a vertical position within 2 s, keeping it that way for 5 s. We then observe whether the rat will come down from the screen or use the front paw to grip the screen to evaluate the gripping ability and muscle strength of the front paw. The scoring standard is divided into four levels: 0: The front paws hold the screen for 5 s for a long time, but will not fall; 1: temporarily hold the screen, slip a distance, but did not fall; 2: fall within 5 s; 3: when the screen turns, the mouse immediately falls [37].

### 4.5. Magnetic Resonance Imaging (MRI)

MRI scans were performed after designated ischemic time or 3 h post drug administration. The animals were anaesthetized with 2.5% isoflurane and fixed in a body restrainer with tooth-bar in an MRI spectrometer (PharmaScan 70/16, Bruker, Germany). Their brains were scanned using a rat head surface coil. Diffusion weighted imaging (DWI) was used with the following parameters: echo time: 22 ms; repetition time: 2500 ms; slice thickness: 0.5 mm; image size: 128 × 128; field of view: 25 × 25 mm; B value: 650 s/mm^2^. Forty successive coronal images were acquired, from which the same position pictures in all the group were selected. Hyperintense infarct areas in DWI images were assigned with a region of interest tool and analyzed using Image-Pro Plus 6.0 software in a blinded manner: infarct area (%) = total infarct area/total section area ×100%; edema ratio (%) = left brain area/right brain area × 100% [66].

### 4.6. ELISA Analysis

After anesthesia, the brain, heart and lung of rats were dissected out. Hippocampus, middle cortex and striatum of the infarcted hemisphere were freed from other parts. The tissues were homogenized with cold physiological saline (1 mg: 9 mL). The homogenate was centrifuged (5000 rpm for 15 min at 4 °C) to collect supernatants for determination of interleukin 1β (IL-1β), tumor necrosis factor α (TNF-α) and lactate dehydrogenase (LDH) activity with ELISA kits (Nanjing Jiancheng Bioengineering Institute, Nanjing, China) according to manufacturer’s instructions in a blinded manner. Quantification of ELISA results was performed at 450 nm in a microplate reader (Thermo Scientific, Waltham, MA, USA).

### 4.7. Nissl Staining

Nissl staining was performed with Cresyl Violet, which can selectively stain the Nissl body in survival neurons [35]. Rats were sacrificed at 3 h post drug administration and then immediately perfused by cold PBS (0.1 M; pH 7.4) and 4% paraformaldehyde. The brain was removed and immersed in fixative. Then, the tissues were embedded in paraffin. Paraffin sections (3 μm) were cut on glass slides, stained with 1% Cresyl Violet dissolved in 0.25% acetic acid, and examined under a light microscope (CKX41, Olympus, Tokyo, Japan) by a pathologist blinded to the study groups (40× and 400×). Cell counts from the left and right hippocampus, cortex, and striatum on each of the six sections were averaged to provide a single value (number of neurons per 200 μm length) for each animal.

### 4.8. Histo-Pathological Examination

Rats were sacrificed and then immediately perfused with cold PBS (0.1 M; pH 7.4) and 4% paraformaldehyde. The apex of the heart, lung were fixed in 4% paraformaldehyde, routinely processed, and embedded in paraffin. Paraffin sections (3 μm) were cut on glass slides, stained with hematoxylin-eosin (HE), and examined under a light microscope (CKX41, Olympus, Tokyo, Japan) by a pathologist blinded to the study groups.

### 4.9. Cardiac Function Detection

The right carotid artery was exposed via a neck dissection. A cannulation was advanced inserted into LV (left ventricle) through right carotid artery of rats, which was connected to a RM6240B multichannel physiological signal acquisition and processing system to record the following parameters of cardiac function, including left ventricular systolic pressure (LVSP), left ventricular developed pressure (LVDP), the maximal ventricular pressure rise ratio during systolic period (+dp/dtmax), and the maximal ventricular pressure decrease ratio during diastolic period (−dp/dtmax).

### 4.10. Pulmonary Function Detection

After endotracheal intubation, a LEAD-7000 multi-channel physiological recorder was used to record the following parameters of pulmonary function, including minute ventilation (VE) and forced expiratory volume in the first second (FEV1).

### 4.11. Immunohistological Staining

Rats were sacrificed and then immediately perfused by cold PBS (0.1 M; pH 7.4) and 4% paraformaldehyde. The brain, apex of the heart, and lung was fixed in 4% paraformaldehyde, routinely processed, and embedded in paraffin. Tissues were cut into 3-μm thick slices. After deparaffinization, endogenous peroxidase was inactivated by 3% H2O2 for 10 min at 37 °C. Then, the slices were permeabilized in PBS for 5 min three times at room temperature. After blocking with 5% bovine serum albumin (BSA) for 1 h, slices were then treated with primary antibody anti-CKLF1 (Peking University Center for Human Disease Genomics, Beijing, China; 1:200 dilution) overnight at 4 °C. After washing three times in PBST (0.1% Tween-20 in PBS), the slices were incubated with biotinylated goat anti-rabbit Abs (074-1506, KPL, Gaithersburg, USA; 1:200 dilution) for 2 h followed by peroxidase-labeled streptavidin (KPL, Gaithersburg, USA; 1:500 dilution) complex for 1 h. The expression of CKLF1 was detected with 3, 3-diaminobenzidine (DBA) as the substrate. Slices were captured using a microscope (CKX41, Olympus, Tokyo, Japan); the numbers of positive cells were scored in ten randomly selected images of the ischemic hemisphere, and the results were expressed as the number of positive cells per image using IMAGE-PRO PLUS 6.0 (Media Cybernetics, MD, USA) in a blinded manner.

### 4.12. Immunofluorescence Staining

The brain, heart and lung were processed for making paraffin section. Slices were immersed in antigen retrieval at over 95 °C for 10 min in a microwave and cooled naturally. After blocking with 5% BSA, the tissues were incubated with primary antibody to CCR4 (ab216560, Abcam, Cambridge, UK; for IF, 1:200 dilution) overnight at 4 °C. Following three washes in PBST (0.1% Tween-20 in PBS), the slices were incubated with a secondary antibody Alexa 488-conjugated donkey anti-rabbit (A21206, Invitrogen, Eugene, OR, USA; 1:500 dilution), and nuclei were stained with Hoechst 33342 (H342, DOJINDO, Kumamoto, Japan; 1:1000 dilution). The number of positive cells was counted in ten randomly selected images under an upright fluorescence microscope (Leica DFC420, Wetzlar, Germany) in a blinded manner, and the results were expressed as the number of positive cells per image (400×).

### 4.13. qPCR

Total RNA was isolated from brain, heart, and lung using TransZol Up kit (TRANSGEN BIOTECH, Beijing, China). RNA was reverse-transcribed to cDNA using TransScript One-Step gDNA Removal (TRANSGEN BIOTECH, Beijing, China) and cDNA Synthesis SuperMix kit (TRANSGEN BIOTECH, Beijing, China). First-strand cDNA was synthesized using 1 μg of total RNA. The amplification of cDNA was performed with an Applied Biosystems 7900HT Fast Real-Time PCR System (Foster City, CA, USA) using TransStart Tip Green qPCR Supermix kit (TRANSGEN BIOTECH, Beijing, China). The mRNA level of individual genes was normalized to the expression of β-actin housekeeping control gene for each sample and calculated using the ΔΔ*C*_T_ method. The primers used for each target gene were as follows, CKLF1 (forward-5′-CGT AGA CCA TCA GCC CTT CTG-3′; reverse-5′-TCA GGA AAC CAA ACA CCC CTC-3′); CCR4 (forward-5′-CAA CGT GGT GCT TTT CCT GG-3′; reverse-5′-CAG GGT TAA GGC AGC AGT GA-3′); IL-1β (forward-5′-AAT GCC TCG TGC TGT CTG A-3′; reverse-5′-AAT GCC TCG TGC TGT CTG A-3′); TNF-α (forward-5′-ACG TCG TAG CAA ACC ACC AA-3′; reverse-5′-GCA GCC TTG TCC CTT GAA GA-3′); actin (forward-5′-TCA GGT CAT CAC TAT CGG CAA T-3′; reverse-5′-AAA GAA AGG GTG TAA AAC GCA-3′).

### 4.14. Western Blotting

Tissues were lysated and proteins were extracted for western blot as described [68]. Tissue were lysed in RIPA lysis buffer (Beyotime, Shanghai, China) for 30 min on ice. After centrifugation at 12,000 rpm for 30 min, the supernatants were collected and protein concentrations were assessed with a BCA kit (Applygen, Beijing, China). Loading buffer was added into the supernatant to denature the protein. Fifty micrograms of proteins from animals in each test group was separated on 15% gels by electrophoresis and then transferred to polyvinylidene difluoride (PVDF) membrane (Millipore, Medford, MA, USA) in blotting buffer for 1 h at 300 mA. The whole PVDF membrane was then blocked with 3% BSA for 2 h at room temperature and then incubated with primary antibodies anti-CKLF1 (Peking University Center for Human Disease Genomics, Beijing, China; 1:500 dilution), anti-CCR4 (ab83250, Abcam, Cambridge, UK; 1:500 dilution), anti-NF-κB-p65 (sc-109, Santa Cruz Biotechnology, Santa Cruz, CA, USA; 1:500 dilution), anti-p-NF-κB-p65 (3036, Cell Signaling Technology, Beverly, MA, USA; 1:500 dilution), anti-IL-1β (ab9722, Abcam, Cambridge, UK; 1:500 dilution), and anti-TNF-α (ab6671, Abcam, Cambridge, UK; 1:500 dilution) at 4 °C overnight. β-actin (ab8226, Abcam, Cambridge, UK; 1:5000 dilution) was measured as loading control. Then secondary antibodies affinity purified antibody peroxidase-labeled goat anti-rabbit IgG H and L (KPL, Gaithersburg, MD, USA) and peroxidase-labeled streptavidin (074-1506, KPL, Gaithersburg, MD, USA; dilution, 1:2000) were incubated for two hours at room temperature with shaking. The expression of each protein was detected with enhanced chemiluminescence (ECL) plus detection system (Molecular Device, Lmax). The density of each band was quantified using image analysis software Gel-Pro Analyzer [69].

### 4.15. Statistical Analysis

All statistical analysis were performed using Prism (7.0; GraphPad Software, La Jolla, CA, USA). All of the data were expressed as the mean ± standard deviation (SD). Data were analyzed using the Student’s t-test or one-way ANOVA followed by the Tukey’s test or were analyzed using two-way ANOVA followed by Bonferroni’s multiple comparison test with Prism 7.00 software. Statistical significance was considered as *p* < 0.05. The number of rats used in each in vivo condition is indicated in the corresponding figure legends.

## 5. Conclusions

In the present study, IMM-H004 was found to have protective effects against permanent focal cerebral ischemia induced brain injury and cardiopulmonary complications in aged rats. CKLF1 is an important mediator showed significant increase after ischemia in brain, heart and lung, and this chemokine is necessary for the IMM-H004 to exert protective effects. IMM-H004 inhibits the expression of CKLF1, suppressing the followed inflammatory response, and further protects the brain, heart, and lung from damage. These findings may have implications for CKLF1 as an important mediator in occurrence of cardiopulmonary complications after ischemic stroke, and also provide new ideas for drug development on ischemic stroke, especially the stroke induced cardiopulmonary complications therapy.

## Figures and Tables

**Figure 1 ijms-20-01661-f001:**
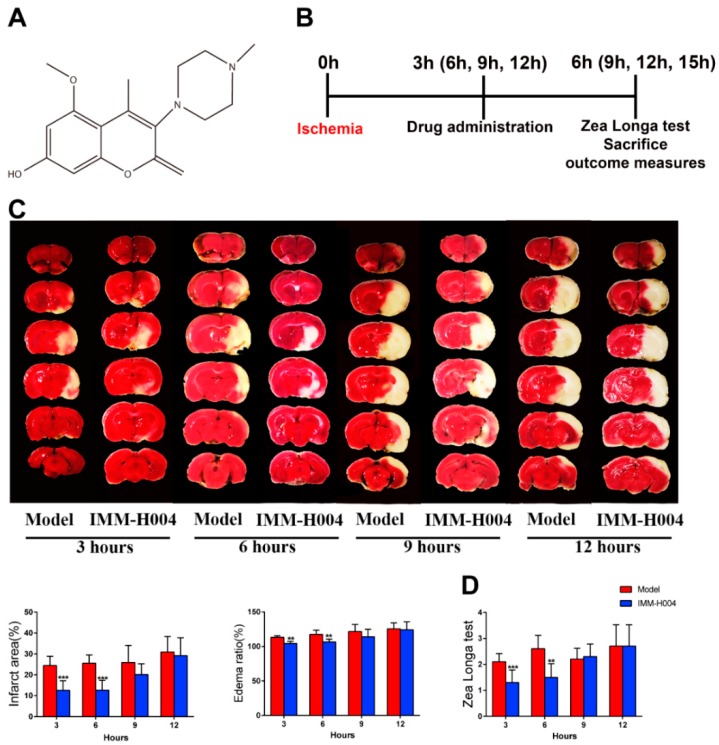
Therapeutic time window experiment of IMM-H004. (**A**) The chemical structure of IMM-H004. (**B**) The timeline diagram of theraputic time window experiment. (**C**) Representative images of TTC staining, and analysis of infarction area and edema ratio. (**D**) Statistical analysis of Zea Longa test scores. Data are shown as the mean ± SD (*n* = 10/group). ** *p* < 0.01 vs. model; *** *p* < 0.001 vs. model.

**Figure 2 ijms-20-01661-f002:**
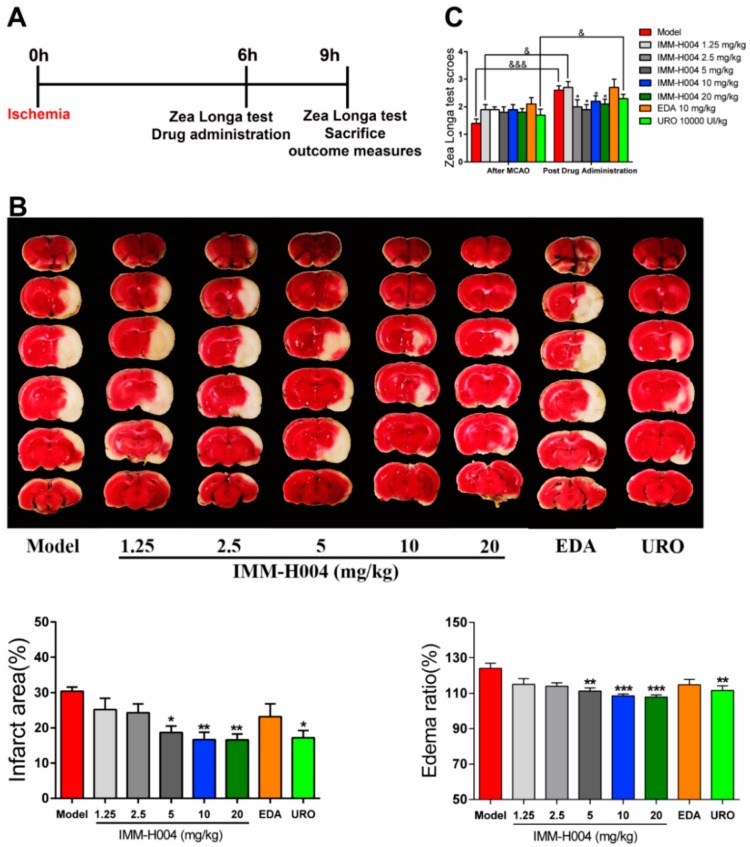
Therapeutic dosage window experiment of IMM-H004. (**A**) The timeline diagram of theraputic dosage window experiment. (**B**) Representative images of TTC staining, and analysis of infarction area and edema ratio. (**C**) Statistical analysis of Zea Longa test scores. Data are shown as the mean ± SD (*n* = 10/group). * *p* < 0.05 vs. model; ** *p* < 0.01 vs. model; *** *p* < 0.001 vs. model; & *p* < 0.05 vs. after MCAO; &&& *p* < 0.001 vs. after MCAO.

**Figure 3 ijms-20-01661-f003:**
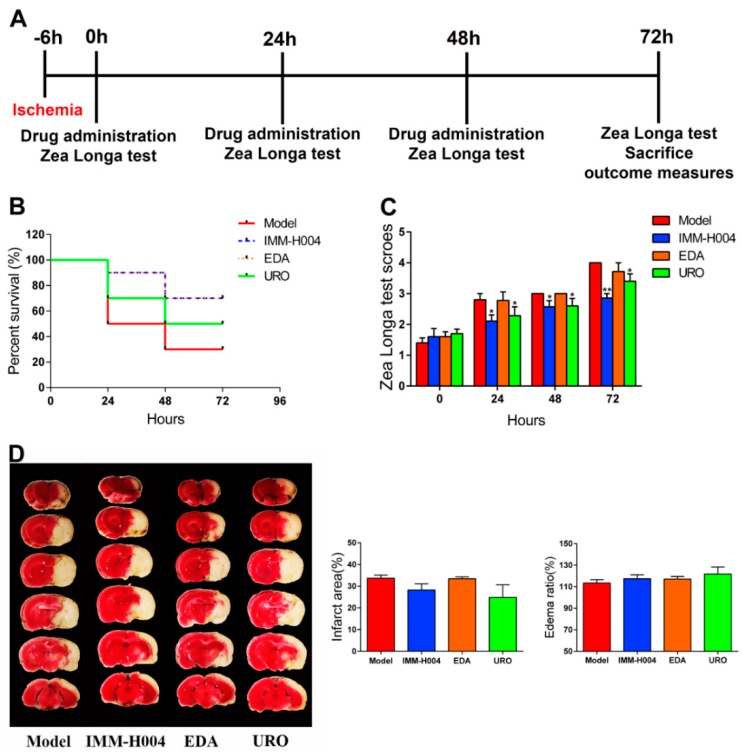
Successive administration of IMM-H004 once daily for three days protects against Permanent focal cerebral ischemia-induced brain injury in adult rats. (**A**) The timeline diagram of successive administration of IMM-H004 once daily for three days. (**B**) Statistical analysis of the 72 h survival rate. (**C**) Statistical analysis of Zea Longa test scores. (**D**) Representative images of TTC staining, and analysis of infarction area and edema ratio. Data are shown as the mean ± SD (*n* = 10/group). * *p* < 0.05 vs. model; ** *p* < 0.01 vs. model.

**Figure 4 ijms-20-01661-f004:**
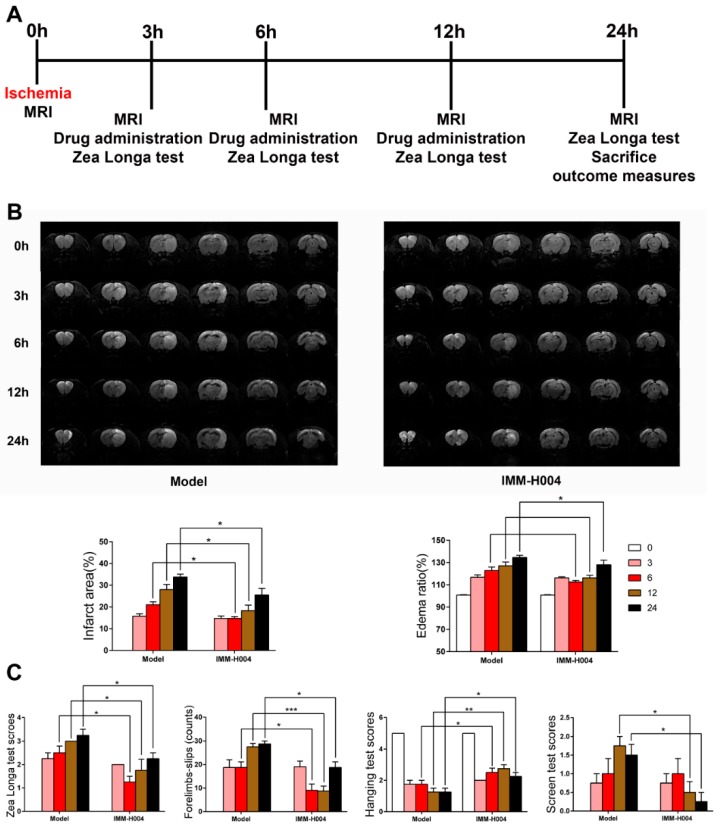
Successive administration of IMM-H004 3 times a day for one day protects against Permanent focal cerebral ischemia-induced brain injury in adult rats. (**A**) The timeline diagram of successive administration of IMM-H004 3 times a day for one day. (**B**) Representative coronal DWI images, and analysis of infarction area and edema ratio. (**C**) Statistical analysis of the Zea Longa test scores, Forelimbs-slips, Hanging test scores, and screen test scores. Data are shown as the mean ± SD (*n* = 10/group). * *p* < 0.05 vs. model; ** *p* < 0.01 vs. model; *** *p* < 0.001 vs. model.

**Figure 5 ijms-20-01661-f005:**
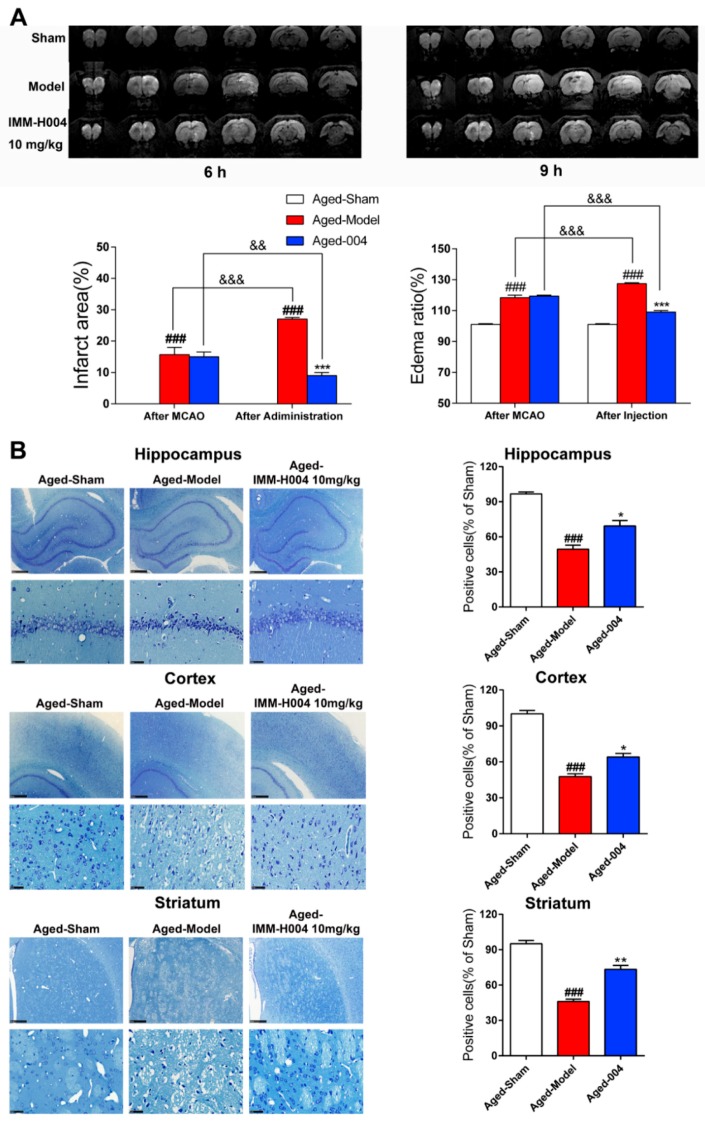
IMM-H004 protects against permanent focal cerebral ischemia-induced brain injury in aged rats. (**A**) Representative coronal DWI images, and analysis of infarction area and edema ratio. (**B**) Representative photographs of Nissl-stained hippocampal CA1, cortex, and striatum subfield from aged rats (50× and 400× magnification, scale bar: 500 and 50 μm) and positive cells ratio. All the data are shown as the mean ± SD; part of data are shown as the mean ± SD after normalization to the sham (*n* = 6/group). ### *p* < 0.001 vs. sham; * *p* < 0.05 vs. model; ** *p* < 0.01 vs. model; *** *p* < 0.001 vs. model; && *p* < 0.01 vs. after MCAO; &&& *p* < 0.001 vs. after MCAO.

**Figure 6 ijms-20-01661-f006:**
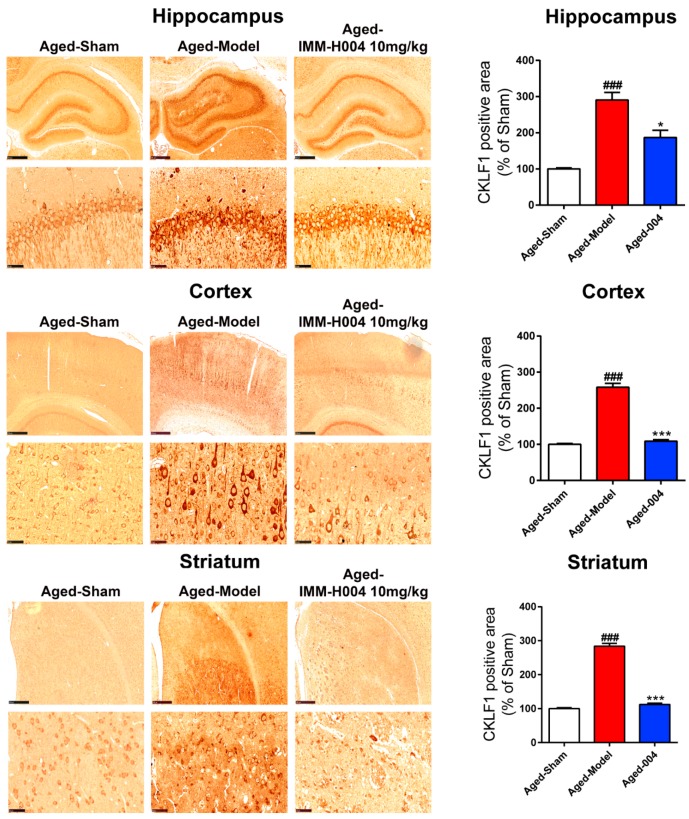
IMM-H004 decreases the expression of CKLF1 in ischemic brain. Expression of CKLF1 in hippocampal CA1, cortex, and striatum subfield from aged rats by immunohistochemical staining (50× and 400× magnification, scale bar: 500 and 50 μm). All the data are shown as the mean ± SD; part of data are shown as the mean ± SD after normalization to the sham (*n* = 6/group). ### *p* < 0.001 vs. sham; * *p* < 0.05 vs. model; *** *p* < 0.001 vs. model.

**Figure 7 ijms-20-01661-f007:**
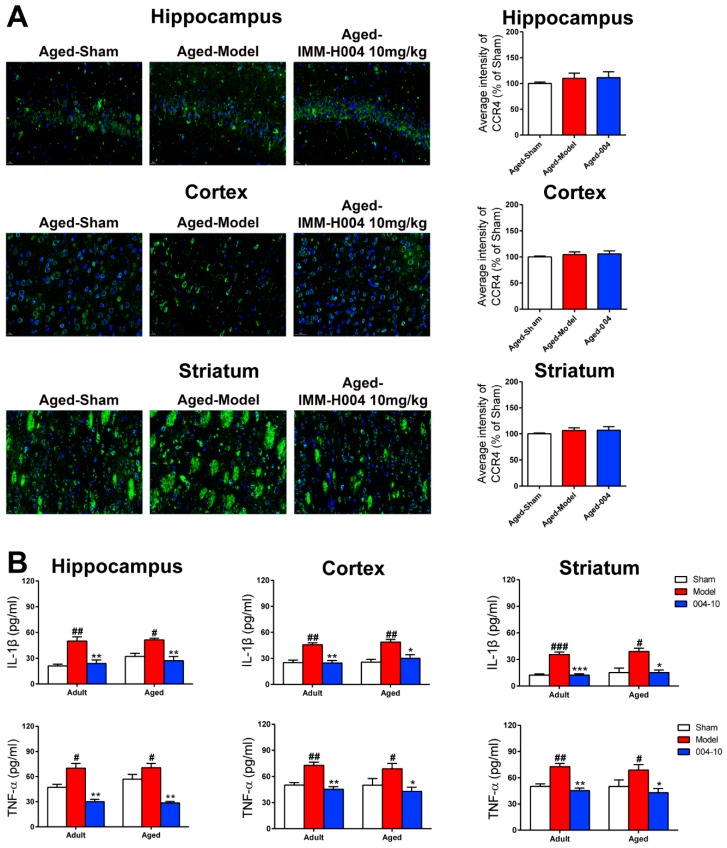
IMM-H004 dereases the inflammatory response in brain of aged rats. (**A**) Expression of CCR4 in hippocampal CA1, cortex, and striatum subfield from aged rats by immunofluorescence staining (400× magnification, scale bar: 50 μm). (**B**) Effect of IMM-H004 on IL-1β and TNF-α levels in hippocampus, cortex, and striatum of aged rats. All the data are shown as the mean ± SD; part of data are shown as the mean ± SD after normalization to the sham (*n* = 6/group). # *p* < 0.05 vs. sham; ## *p* < 0.01 vs. sham; ### *p* < 0.001 vs. sham; * *p* < 0.05 vs. model; ** *p* < 0.01 vs. model; *** *p* < 0.001 vs. model.

**Figure 8 ijms-20-01661-f008:**
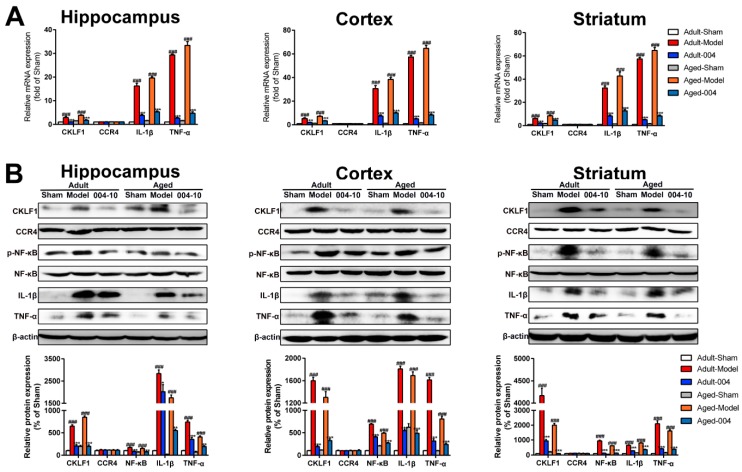
IMM-H004 decreases Permanent focal cerebral ischemia-induced brain inflammation in aged rats. (**A**) RT-PCR analysis for mRNA levels of CKLF1, CCR4, IL-1β, and TNF-α in hippocampus, cortex, and striatum of aged rats. (**B**) Representative blots and densitometry data for CKLF1, CCR4, IL-1β, and TNF-α hippocampus, cortex, and striatum of aged rats. Data are shown as the mean ± SD after normalization to the sham (*n* = 6/group). ### *p* < 0.001 vs. sham; ** *p* < 0.01 vs. model; *** *p* < 0.001 vs. model.

**Figure 9 ijms-20-01661-f009:**
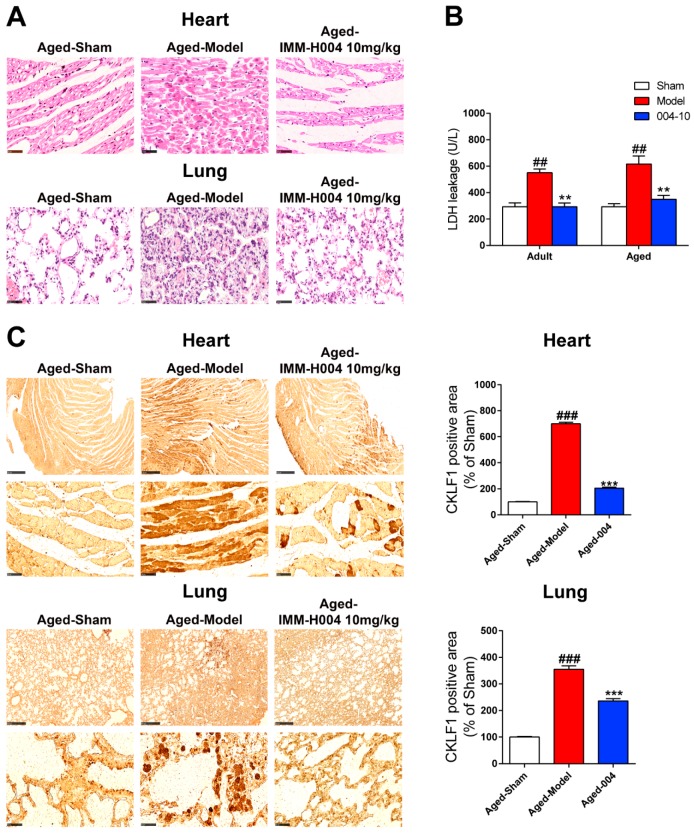
IMM-H004 protects against Permanent focal cerebral ischemia-induced cardiopulmonary complications in aged rats. (**A**) Representative photographs of HE-stained heart and lung subfield from aged rats (400× magnification, scale bar: 50 μm). (**B**) Effect of IMM-H004 on LDH level in heart of aged rats using an LDH assay kit. (**C**) Expression of CKLF1 in heart and lung subfield from aged rats by immunohistochemical staining (50× and 400× magnification, scale bar: 500 and 50 μm). All the data are shown as the mean ± SD; part of data are shown as the mean ± SD after normalization to the sham (*n* = 6/group). ## *p* < 0.01 vs. sham; ### *p* < 0.001 vs. sham; ** *p* < 0.01 vs. model; *** *p* < 0.001 vs. model.

**Figure 10 ijms-20-01661-f010:**
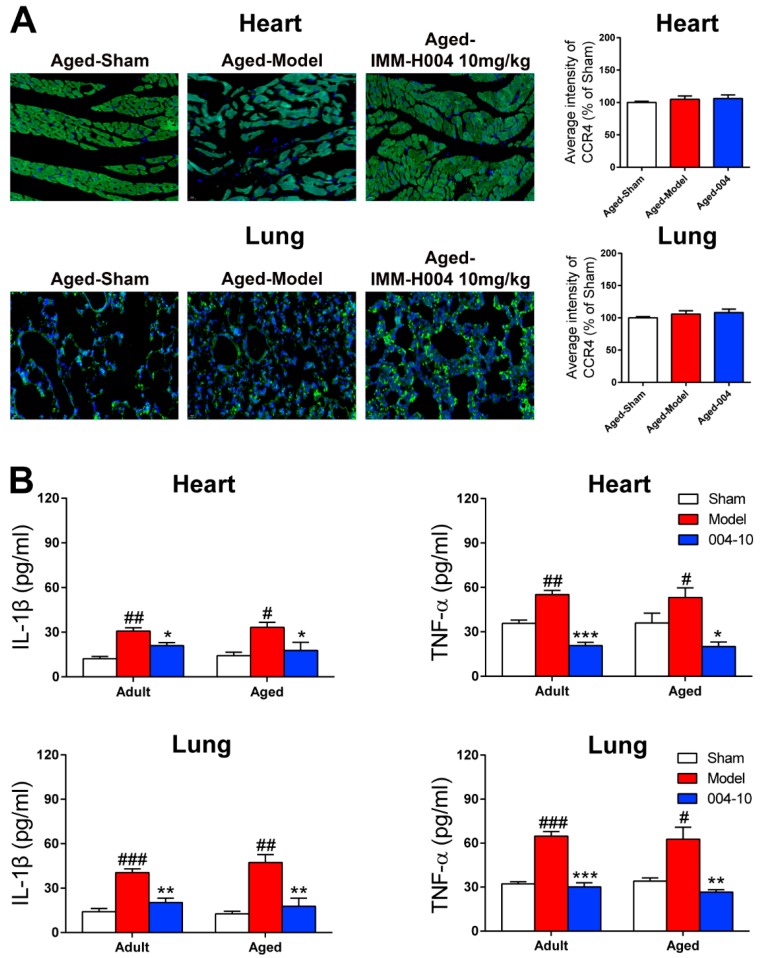
IMM-H004 decreases inflammation in heart and lung of aged rats. (**A**) Expression of CCR4 in heart and lung subfield from aged rats by immunofluorescence staining (400× magnification, scale bar: 50 μm). (**B**) Effect of IMM-H004 on IL-1β and TNF-α levels in aged rats heart and lung using IL-1β and TNF-α assay kits. All the data are shown as the mean ± SD; part of data are shown as the mean ± SD after normalization to the sham (*n* = 6/group). # *p* < 0.05 vs. sham; ## *p* < 0.01 vs. sham; ### *p* < 0.001 vs. sham; * *p* < 0.05 vs. model; ** *p* < 0.01 vs. model; *** *p* < 0.001 vs. model.

**Figure 11 ijms-20-01661-f011:**
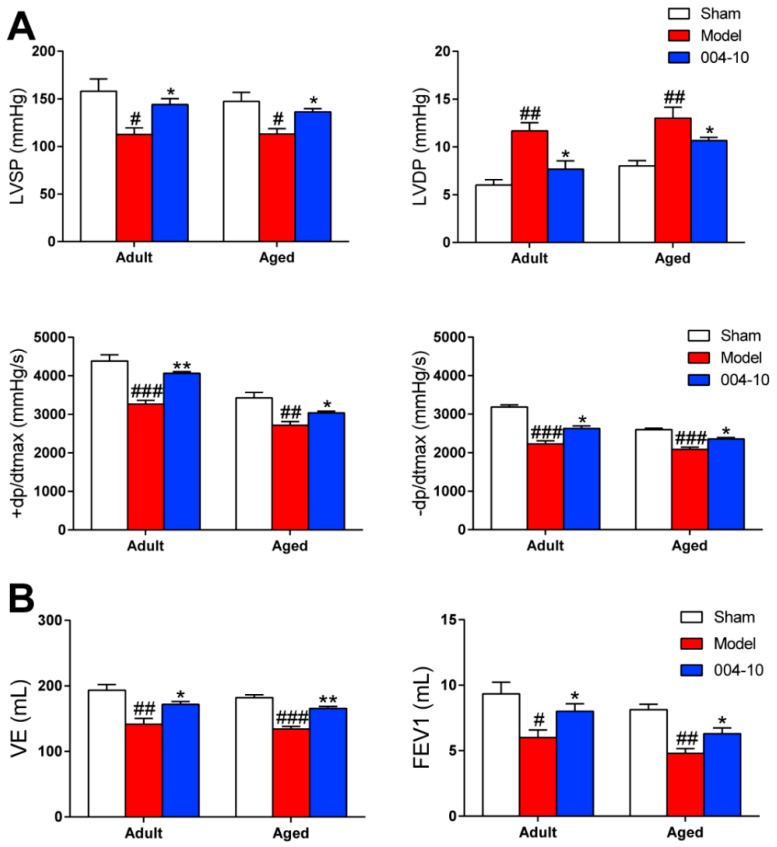
IMM-H004 protects cardiopulmonary function in adult and aged rats. (**A**) Effect of IMM-H004 on adult and aged rats cardiac function including the detection of LVSP, LVDP, +dp/dtmax, and -dp/dtmax. (**B**) Effect of IMM-H004 on adult and aged rats pulmonary function including the detection of VE and FEV1. All the data are shown as the mean ± SD (*n* = 6/group). # *p* < 0.05 vs. sham; ## *p* < 0.01 vs. sham; ### *p* < 0.001 vs. sham; * *p* < 0.05 vs. model; ** *p* < 0.01 vs. model.

**Figure 12 ijms-20-01661-f012:**
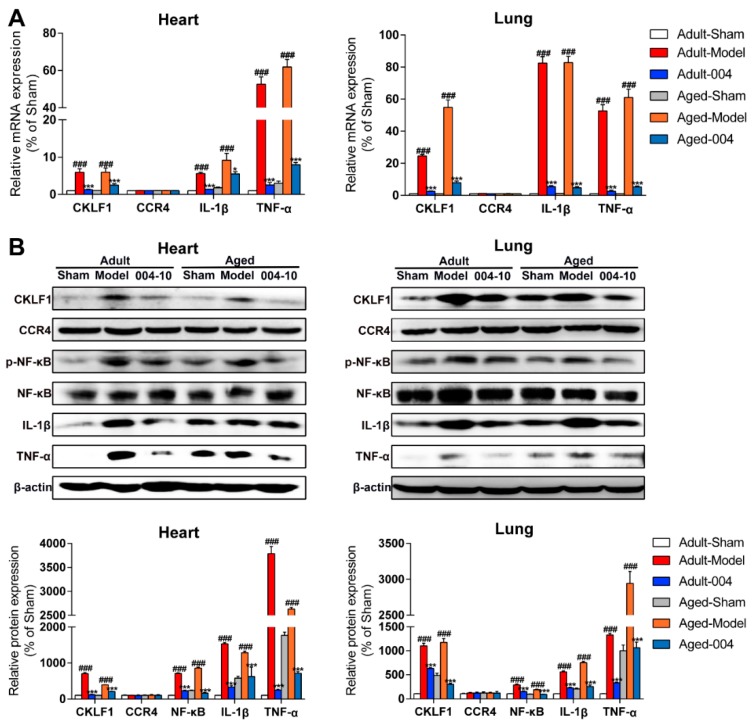
IMM-H004 decreases inflammation in heart and lung of aged rats. (**A**) Effect of IMM-H004 on aged rats heart and lung mRNA levels of CKLF1, CCR4, IL-1β, and TNF-α as determined by quantitative RT-PCR. (**B**) Effect of IMM-H004 on the CKLF1, CCR4, IL-1β, and TNF-α protein expression levels in aged rats heart and lung as assayed by Western blotting using a Gel-Pro analyzer (Media Cybernetics, Rockville, MD, USA). All the data are shown as the mean ± SD; part of data are shown as the mean ± SD after normalization to the sham (*n* = 6/group). ### *p* < 0.001 vs. sham; * *p* < 0.05 vs. model; *** *p* < 0.001 vs. model.

**Figure 13 ijms-20-01661-f013:**
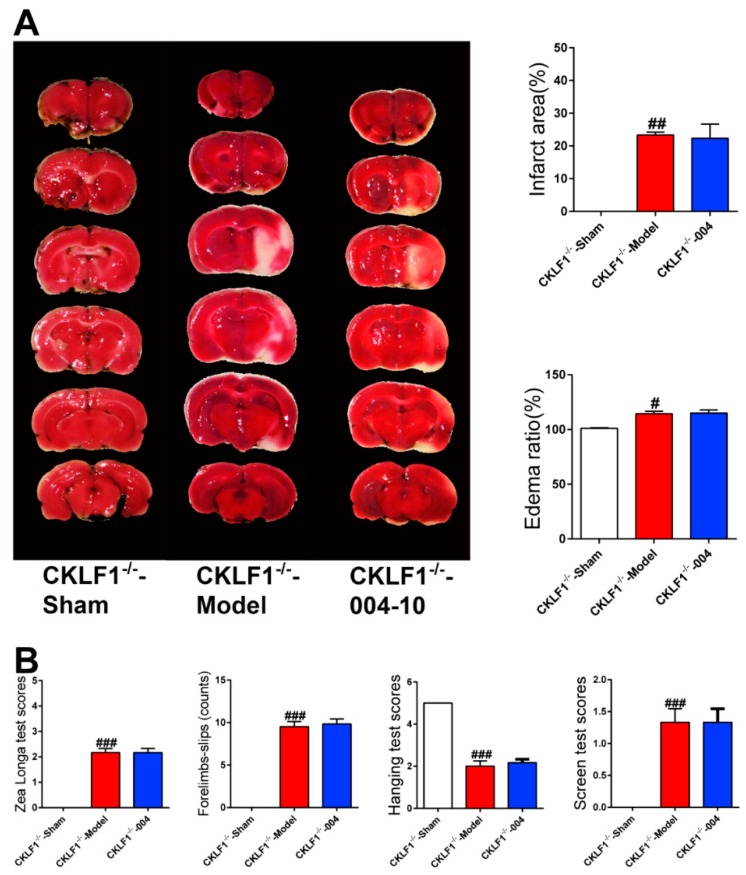
CKLF1 is necessary for IMM-H004 to exert protective effects against brain ischemia. (**A**) Representative TTC staining images, and analysis of infarction area and edema ratio. (**B**) Statistical analysis of the Zea Longa test scores, forelimb slips, hanging test scores, and screen test scores in CKLF1^−/−^ rats. All the data are shown as the mean ± SD, a part of data are shown as the mean ± SD after normalization to the sham (*n* = 6/group). # *p* < 0.05 vs. sham; ## *p* < 0.01 vs. sham; ### *p* < 0.001 vs. sham.

**Figure 14 ijms-20-01661-f014:**
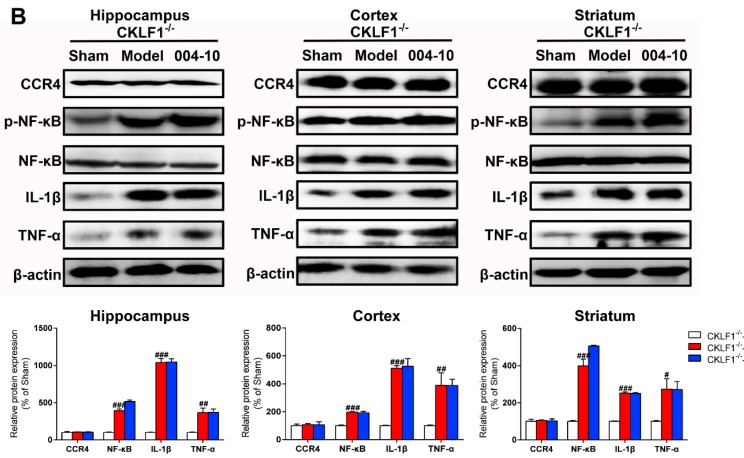
CKLF1 is necessary for IMM-H004 to anti-inflammation in ischemic brain. (**A**) RT-PCR analysis for Effect of IMM-H004 on expression levels of CCR4, IL-1β, and TNF-α in the hippocampus, cortex, and striatum of CKLF1^−/−^ rats. (**B**) Western blot analysis for Effect of IMM-H004 on expression of CCR4, p-NF-κB, NF-κB, IL-1β, and TNF-α in the hippocampus, cortex, and striatum of CKLF1^−/−^ rats. All the data are shown as the mean ± SD, a part of data are shown as the mean ± SD after normalization to the sham (*n* = 6/group). # *p* < 0.05 vs. sham; ## *p* < 0.01 vs. sham; ### *p* < 0.001 vs. sham.

**Figure 15 ijms-20-01661-f015:**
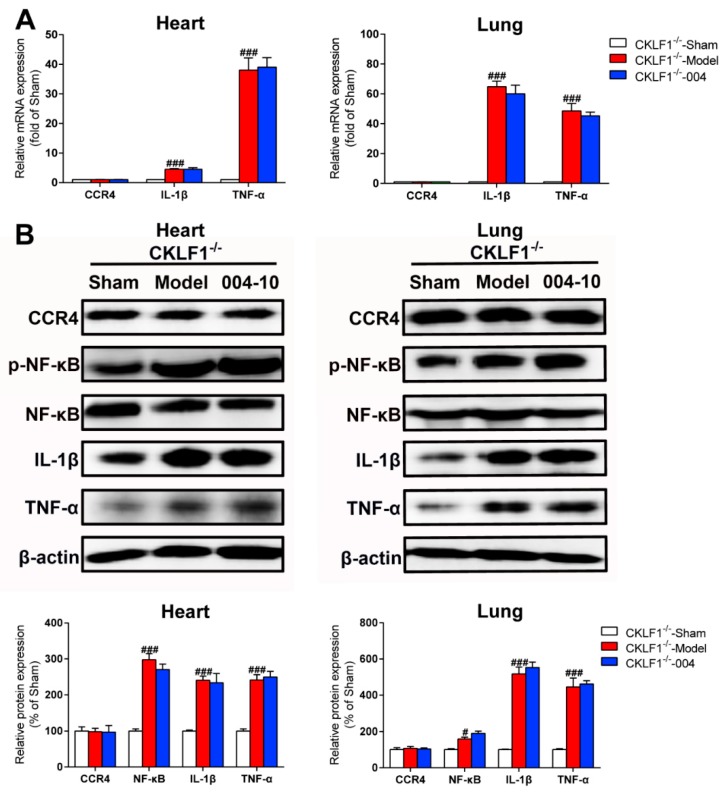
CKLF1 is necessary for IMM-H004 to suppress inflammation in heart and lung. (**A**) RT-PCR analysis for Effect of IMM-H004 on CCR4, p-NF-κB, NF-κB, IL-1β, and TNF-α in the heart and lung of CKLF1^−/−^ rats. (**B**) Western blot analysis for the effect of IMM-H004 on CCR4, IL-1β, and TNF-α in heart and lung of CKLF1^−/−^ rats. All the data are shown as the mean ± SD, a part of data are shown as the mean ± SD after normalization to the sham (*n* = 6/group). # *p* < 0.05 vs. sham; ### *p* < 0.001 vs. sham.

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
