# Peer review of "IMM-H004 Protects against Cerebral Ischemia Injury and Cardiopulmonary Complications via CKLF1 Mediated Inflammation Pathway in Adult and Aged Rats"

_ijms, 2019, doi:10.3390/ijms20071661_

Round 1

Reviewer 1 Report

Effects of IMM-H004 on cerebral ischemia-induced changes in histological and biochemical parameters in brain, heart, and lung were examined.  The authors showed that IMM-H004-treatment of the animals with cerebral ischemia improved these parameters after an exposure to ischemia.  This study is interesting.  However, there are unclear points in this study.

Comments;

1.  There are no data about changes in histological, behavioral, and biochemical parameters of normal and sham-operated animals in this study.  They are very important data for the present study.

2.  In this study, although the authors showed effects of IMM-H004 on changes in histological and biochemical parameters in heart and lung, there are no data about effects of the agent on these tissue functions.  Therefore, cardiac and pulmonary functional parameters of the animals treated with and without the agent should be shown.

Author Response

Dear reviewer:

I deeply appreciate for the comments concerning our manuscript entitled “IMM-H004 Protects Against Cerebral Ischemia Injury and Cardiopulmonary Complications via CKLF1 mediated Inflammation Pathway in Adult and Aged Rats”. Those comments are all valuable and very helpful for revising and improving our paper, as well as the important guiding significance to our researches. We have studied comments carefully and have made corrections which we hope meet with approval. The main corrections in the paper and the responds to the your comments are as following:

Response to Reviewer 1 Comments

Comment 1: There are no data about changes in histological, behavioral, and biochemical parameters of normal and sham-operated animals in this study. They are very important data for the present study.

Response 1: Thank you very much for your comment.This comment is important. Usually we use sham-operated group instead of normal group in animal study [1, 2]. Animals will use an anesthesia machine for deep anesthesia before the experiment to eliminate the stress response caused by sham surgery. We have conducted a lot of preliminary experiments in the early stage found that there are no changes between normal and sham-operated group in histological, behavioral, and biochemical parameters. So we use sham-operated group instead of normal group in animal study.

Comment 2: In this study, although the authors showed effects of IMM-H004 on changes in histological and biochemical parameters in heart and lung, there are no data about effects of the agent on these tissue functions. Therefore, cardiac and pulmonary functional parameters of the animals treated with and without the agent should be shown.

Response 2: Thank you very much for your comment. This comment is very important. We have tested the cardiopulmonary function in the early stage, now I will provide it to you. The right carotid artery was exposed via a neck dissection. A cannulation was advanced inserted into LV (left ventricle) through right carotid artery of rats, which was connected to a RM6240B multichannel physiological signal acquisition and processing system to record the following parameters of cardiac function, including Left ventricular systolic pressure (LVSP), Left ventricular developed pressure (LVDP), The maximal ventricular pressure rise ratio during systolic period (+dp/dtmax) and The maximal ventricular pressure decrease ratio during diastolic period (−dp/dtmax). After endotracheal intubation, LEAD-7000 Multi-channel Physiological Recorder was used to record the following parameters of pulmonary function, including Minute ventilation (VE) and Forced expiratory volume in the first second (FEV1).

Figure 11. IMM-H004 protects cardiopulmonary function in adult and aged rats. (A) Effect of IMM-H004 on adult and aged rats cardiac function including the detection of LVSP, LVDP, +dp/dtmax, and -dp/dtmax. (B) Effect of IMM-H004 on adult and aged rats pulmonary function including the detection of VE and FEV1. All the data are shown as the mean ± SD (n = 6/group). #p < 0.05 vs. sham; ##p < 0.01 vs. sham; ###p < 0.001 vs. sham;*p < 0.05 vs. model; **p < 0.01 vs. model.

1.      Chu, S.F.; Zhang, Z.; Zhang, W.; Zhang, M.J.; Gao, Y.; Han, N.; Zuo, W.; Huang, H.Y. ; Chen, N.H. Upregulating the Expression of Survivin-HBXIP Complex Contributes to the Protective Role of IMM-H004 in Transient Global Cerebral Ischemia/Reperfusion. Mol Neurobiol. 2017, 54, 524-540.

2.      Liu, S.; Ai, Q.; Feng, K.; Li, Y. ; Liu, X. The cardioprotective effect of dihydromyricetin prevents ischemia-reperfusion-induced apoptosis in vivo and in vitro via the PI3K/Akt and HIF-1alpha signaling pathways. Apoptosis. 2016, 21, 1366-1385.

We deeply thank again for the prudent comments on our manuscript.

Very sincerely yours,

Qidi Ai

Reviewer 2 Report

This study aims to examine the capacity of IMM-H004 in inhibiting CKLF1-CCR4 interaction in reducing brain injury and other extra-cerebral complications after stroke. While the manuscript is well written, and clearly this is an important area of research for such prevalent cerebral disease, unfortunately some of the conclusions made in the study are not supported by the results. One of the glaring issue is the claim that this study examined cardiopulmonary complications. This is not true, merely examining histological sections do not reflect organ function nor post-stroke infection. Other concerns:

1) While this reviewer appreciates the testing of multiple timepoints for therapeutic window of IMM-H004, it is lost to me why the experimental endpoint is only 3h after IMM-H004 administration? Is IMM-H004 bioactive after 3h? Would a better experimental design to inject the IMM-H004 at 3, 6, 9, or 12 h post-stroke and examine the infarct 24 h after stroke onset? 

2) The experimental timepoint and endopoints of Fig 2 is not stated. 

3) It is quite alarming that only 50% of the post-stroke rats survive this model. Despite I agree that IMM-H004 may be effective in reducing mortality and improve behaviour, the data in Fig 3D does not support the authors claim that IMM-H004 has a protective effect on brain injury (line 132). This needs to be rectify.

4) Fig 4 supports the idea that IMM-H004 has a very short bioactive period. The potential neuroprotective effect is only apparent when multiple administration is made within 24h - this needs to be highlighted and discussed.

5) Neurons number do not increase (line 171), do you mean "protected from stroke-induced reduction"?

6) Data presentation of Fig 6 came after Fig 7, please rectify.

7) It is very surprising that the inflammatory signature of aged and adult rats in this study seem identical, while numerous studies have eluded to a chronic pro-inflammatory state in aging. Please explain.

8) The authors claim of "rats suffered with pMCAO operation... characterized by inflammatory cell infiltration, alveolar wall thickening, pulmonary interstitial edema, and hemorrhage) - none of these measures were actually quantified. Clearly, post-stroke cardiac fibrosis and pulmonary infection need to be assessed and shown to reduce if these are the complications the authors propose IMM-H004 will reduce. Moreover, the timepoint at which the heart and lung were assessed following stroke onset was not clear - days or weeks after?

9) Animal ethics info is missing.

Author Response

Dear reviewer:

I deeply appreciate for the comments concerning our manuscript entitled “IMM-H004 Protects Against Cerebral Ischemia Injury and Cardiopulmonary Complications via CKLF1 mediated Inflammation Pathway in Adult and Aged Rats”. Those comments are all valuable and very helpful for revising and improving our paper, as well as the important guiding significance to our researches. We have studied comments carefully and have made corrections which we hope meet with approval. The main corrections in the paper and the responds to the your comments are as following:

Response to Reviewer 2 Comments

This study aims to examine the capacity of IMM-H004 in inhibiting CKLF1-CCR4 interaction in reducing brain injury and other extra-cerebral complications after stroke. While the manuscript is well written, and clearly this is an important area of research for such prevalent cerebral disease, unfortunately some of the conclusions made in the study are not supported by the results. One of the glaring issue is the claim that this study examined cardiopulmonary complications. This is not true, merely examining histological sections do not reflect organ function nor post-stroke infection.

Response: Thank you very much for your comment. This comment is so important and helpful for the further research.

1. Merely examining histological sections do not reflect organ function nor post-stroke infection. We have tested the cardiopulmonary function in the early stage, now I will provide it to you. The right carotid artery was exposed via a neck dissection. A cannulation was advanced inserted into LV (left ventricle) through right carotid artery of rats, which was connected to a RM6240B multichannel physiological signal acquisition and processing system to record the following parameters of cardiac function, including Left ventricular systolic pressure (LVSP), Left ventricular developed pressure (LVDP), The maximal ventricular pressure rise ratio during systolic period (+dp/dtmax) and The maximal ventricular pressure decrease ratio during diastolic period (−dp/dtmax). After endotracheal intubation, LEAD-7000 Multi-channel Physiological Recorder was used to record the following parameters of pulmonary function, including Minute ventilation (VE) and Forced expiratory volume in the first second (FEV1). I think it can support my conclusion.

2. As for the post-stroke infection, inflammation has been detected in the organ. Because of the early complications of our detection, the observation time is 9 h after ischemia, and the infection may not be very serious. We will observe the relevant indicators in the further research.

Figure 11. IMM-H004 protects cardiopulmonary function in adult and aged rats. (A) Effect of IMM-H004 on adult and aged rats cardiac function including the detection of LVSP, LVDP, +dp/dtmax, and -dp/dtmax. (B) Effect of IMM-H004 on adult and aged rats pulmonary function including the detection of VE and FEV1. All the data are shown as the mean ± SD (n = 6/group). #p < 0.05 vs. sham; ##p < 0.01 vs. sham; ###p < 0.001 vs. sham;*p < 0.05 vs. model; **p < 0.01 vs. model.

Comment 1: While this reviewer appreciates the testing of multiple timepoints for therapeutic window of IMM-H004, it is lost to me why the experimental endpoint is only 3 h after IMM-H004 administration? Is IMM-H004 bioactive after 3 h? Would a better experimental design to inject the IMM-H004 at 3, 6, 9, or 12 h post-stroke and examine the infarct 24 h after stroke onset?

Response 1: Thank you very much for your comment.This comment is prudent. We inject the IMM-H004 at 3, 6, 9, 12, and 24 h post-stroke and examine the infarct 3, 6 and 9 h after IMM-H004 administration in the previous therapeutic time window study. The results showed that compared with the pMCAO-operated group, rats administered with IMM-H004 (10 mg/kg) showed smaller brain infarct size and ameliorated neurological deficits pronouncedly at 3 and 6 h after ischemia, but IMM-H004 administration showed no protective effect at 9, 12 and 24 h after ischemia. And only 3 h after IMM-H004 administration has the protective effect. These results was provided in Fig 1. So the therapeutic time window of IMM-H004 for ischemia is 0 to 6 h.

Figure 1. Therapeutic time window experiment of IMM-H004. (A) The chemical structure of IMM-H004. (B) The timeline diagram of theraputic time window experiment. (C) Representative images of TTC staining, and analysis of infarction area and edema ratio. (D) Statistical analysis of Zea Longa test scores. Data are shown as the mean ± SD (n = 10/group). **p < 0.01 vs. model; ***p < 0.001 vs. model.

Comment 2: The experimental timepoint and endopoints of Fig 2 is not stated.

Response 2: Thank you very much for your comment.This comment is prudent. As described in Fig 2A, the ischemia time is 6 h. We inject IMM-H004 in 6 h and sacrifice animals 3 h after administration in order to determine the therapeutic dosage window of IMM-H004. Zea Longa test was executed in 6 h and 9 h after pMCAO. I have modified it in line 111-112 in the paper yet.

Figure 2. Therapeutic dosage window experiment of IMM-H004. (A) The timeline diagram of theraputic dosage window experiment. (B) Representative images of TTC staining, and analysis of infarction area and edema ratio. (C) Statistical analysis of Zea Longa test scores. Data are shown as the mean ± SD (n = 10/group). *p < 0.05 vs. model; **p < 0.01 vs. model; ***p < 0.001 vs. model; &p < 0.05 vs. after MCAO; &&&p < 0.001 vs. after MCAO.

Comment 3: It is quite alarming that only 50% of the post-stroke rats survive this model. Despite I agree that IMM-H004 may be effective in reducing mortality and improve behaviour, the data in Fig 3D does not support the authors claim that IMM-H004 has a protective effect on brain injury (line 132). This needs to be rectify.

Response 3: Thank you very much for your comment.This comment is so prudent. It may be due to the long time of ischemia and the low number of doses, resulting in low survival rate of rats. The results showed that that continuous administration of IMM-H004 (10 mg/kg) has a better improvement effect than Edaravone and Urokinase on permanent focal cerebral ischemia induced brain injury in rats beyond 6 h treatment time window. I have rectified it in line 133-134 in the paper yet.

Comment 4: Fig 4 supports the idea that IMM-H004 has a very short bioactive period. The potential neuroprotective effect is only apparent when multiple administration is made within 24 h - this needs to be highlighted and discussed.

Response 4: Thank you very much for your comment.This comment is so prudent and helpful for the further research. Thrombolytic therapy is commonly used in patients with acute ischemic stroke, and recombinant tissue plasminogen activator (tPA) is the only Food and Drug Administration (FDA)-approved drug in clinical application. It is widely known that thrombolytic therapy only beneficial within 4.5 h after ischemic onset. Fig 1 showed that the therapeutic time window of IMM-H004 in single administration on permanent focal cerebral ischemia was determined to be 0-6 h which is wider than that of tPA. In order to extend the therapeutic time window, multiple drug administration daily was used. Fig 4 showed that IMM-H004 (10 mg/kg) multiple dosing administration also pronouncedly reduced the brain infarct size and attenuated neurological deficits. Because the protection mechanism of IMM-H004 is not thrombolytic therapy but anti-inflammation, so multiple dosing administration still have protective effect in long time beyond 6 h. In the further research we will investigate the effects of multiple administrations for a longer period of time beyond 24 h.

Figure 4. Successive administration of IMM-H004 3 times a day for one day protects against Permanent focal cerebral ischemia-induced brain injury in adult rats. (A) The timeline diagram of successive administration of IMM-H004 3 times a day for one day. (B) Representative coronal DWI images, and analysis of infarction area and edema ratio. (C) Statistical analysis of the Zea Longa test scores, Forelimbs-slips, Hanging test scores, and Screen test scores. Data are shown as the mean ± SD (n = 10/group). *p < 0.05 vs. model; **p < 0.01 vs. model; ***p < 0.001 vs. model.

Comment 5: Neurons number do not increase (line 171), do you mean "protected from stroke-induced reduction"?

Response 5: Thank you very much for your comment.This comment is prudent. What I mean is "protected from stroke-induced reduction". I have modified it in line 175 in the paper yet.

Comment 6: Data presentation of Fig 6 came after Fig 7, please rectify.

Response 6: Thank you very much for your comment.This comment is prudent. I have modified it in line 177-194 in the paper yet.

Comment 7: It is very surprising that the inflammatory signature of aged and adult rats in this study seem identical, while numerous studies have eluded to a chronic pro-inflammatory state in aging. Please explain.

Response 7: Thank you very much for your comment.This comment is so prudent and helpful for the further research. The results showed that the inflammatory signature of aged and adult rats in this study seem identical. May be caused by the following reasons: 1. This may be related to the tolerance of different age animals to cerebral ischemia. Different degrees of tolerance to cerebral ischemia may result in no difference in the extent of their inflammatory response. In the further research we will investigate it.

2. May be related to modeling. Unlike adult rats, the weight of the aged rats is not uniform (from 500-600g), causing differences in the degree of nylon filament obstruction, resulting in differences in damage. We will try to eliminate this effect in the further research.

Comment 8: The authors claim of "rats suffered with pMCAO operation... characterized by inflammatory cell infiltration, alveolar wall thickening, pulmonary interstitial edema, and hemorrhage) - none of these measures were actually quantified. Clearly, post-stroke cardiac fibrosis and pulmonary infection need to be assessed and shown to reduce if these are the complications the authors propose IMM-H004 will reduce. Moreover, the timepoint at which the heart and lung were assessed following stroke onset was not clear - days or weeks after?

Response 8: Thank you very much for your comment.This comment is so prudent and helpful for the further research.

1. Some biochemical parameters of cardiac and pulmonary and the cardiopulmonary function detection data were used to show the cardiopulmonary complication by ischemia stroke. These measures were actually quantified.

2. As for the post-stroke cardiac fibrosis and pulmonary infection, because of the early complications of our detection, the observation time is 9 h after ischemia, and the infection may not be very serious. We will observe the relevant indicators in the further research.

3. The timepoint at which the heart and lung were assessed is 9 h after pMCAO. In the further research we will investigate days or weeks after pMCAO to determine whether IMM-H004 still have protective effect in cardiopulmonary complications by cerebral ischemia injury.

Comment 9: Animal ethics info is missing.

Response 9: Thank you very much for your comment.This comment is so prudent. I have provided it in line 414 in the paper.

We deeply thank again for the prudent comments on our manuscript.

Very sincerely yours,

Qidi Ai

Round 2

Reviewer 1 Report

The manuscript has been improved.  However, there are no data about changes in parameters of the sham-operated animals treated with IMM-H004.

Author Response

Dear reviewer:

I deeply appreciate for the comments concerning our manuscript entitled “IMM-H004 Protects Against Cerebral Ischemia Injury and Cardiopulmonary Complications via CKLF1 mediated Inflammation Pathway in Adult and Aged Rats”. Those comments are all valuable and very helpful for revising and improving our paper, as well as the important guiding significance to our researches. We have studied comments carefully and have made corrections which we hope meet with approval. The main corrections in the paper and the responds to the your comments are as following:

Response to Reviewer 1 Comments

Comment 1: The manuscript has been improved. However, there are no data about changes in parameters of the sham-operated animals treated with IMM-H004.

Response 1: Thank you very much for your comment.This comment is very important. We are sorry for misunderstanding this question in the round 1 revision. Actually, in the pre-experiment, we set the group of sham-004 (sham-operated rats treated with IMM-H004), and found that there were no significant differences between the sham group and sham-004 group in parameters we checked (infarction by TTC staining, neurobehavioral deficit tests, and inflammatory cytokines by ELISA analysis). We added these data about changes in parameters of the sham-operated rats treated with IMM-H004 in the supplementary data.

Supplementary Figure 1. Effect of IMM-H004 on sham group rats. (A) Representative images of TTC staining, and analysis of infarction area and edema ratio. (B) Statistical analysis of the Zea Longa test scores, Forelimbs-slips, Hanging test scores, and Screen test scores. Data are shown as the mean ± SD (n = 10/group).

Supplementary Figure 2. Effect of IMM-H004 on sham group rats. (A) Effect of IMM-H004 on IL-1β and TNF-α levels in hippocampus, cortex, striatum, heart, and lung of adult rats. (B) Effect of IMM-H004 on LDH level in heart of adult rats using an LDH assay kit. Data are shown as the mean ± SD (n = 6/group).

We deeply thank again for the prudent comments on our manuscript.

Very sincerely yours,

Qidi Ai

Reviewer 2 Report

The manuscript has improved by the inclusion of Figure 11, showing parameters of cardiopulmonary complications. However, although the authors addressed the reviewers' concerns in their response document, the manuscript has not been modified to reflect these concerns. Namely:

Line 132 - I still do not agree with your conclusion of this experiment, please amend to read "These results demonstrated that continuous administration of IMM-H004 (10 mg/kg) showed improved survival compared to Edaravone and Urokinase after permanent focal cerebral ischemia in rats, however the infarct size remains unchanged (Figure 3B-D)."

Line 174-175 - please amend to read "IMM-H004 significantly protected the number of neurons in these brain regions from the stroke-induced reduction (Figure 5B).

Line 264-267 and Figure 11 - define the acronyms and parameters you are measuring for readers not in the field. Mention that this was done at what timepoint post-stroke and post-drug. Mentioned lung infection and inflammation were not found.

Related to Comment 4 - modify the manuscript to include your response in the text. I specify this response "needs to be highlighted and discussed".

Author Response

Dear reviewer:

I deeply appreciate for the comments concerning our manuscript entitled “IMM-H004 Protects Against Cerebral Ischemia Injury and Cardiopulmonary Complications via CKLF1 mediated Inflammation Pathway in Adult and Aged Rats”. Those comments are all valuable and very helpful for revising and improving our paper, as well as the important guiding significance to our researches. We have studied comments carefully and have made corrections which we hope meet with approval. The main corrections in the paper and the responds to the your comments are as following:

Response to Reviewer 2 Comments

The manuscript has improved by the inclusion of Figure 11, showing parameters of cardiopulmonary complications. However, although the authors addressed the reviewers' concerns in their response document, the manuscript has not been modified to reflect these concerns. Namely:

Response: Thank you for your precious comments, and we are sorry that the manuscript has not been modified to reflect these concerns. We modified the manuscript according to your advices carefully and hoped to resolve these problems.

Comment 1: I still do not agree with your conclusion of this experiment, please amend to read "These results demonstrated that continuous administration of IMM-H004 (10 mg/kg) showed improved survival compared to Edaravone and Urokinase after permanent focal cerebral ischemia in rats, however the infarct size remains unchanged (Figure 3B-D)."

Response 1: Thank you very much for your comment.This comment is so prudent. We have amended the substance in line 132 “These results demonstrated that continuous administration of IMM-H004 (10 mg/kg) has a protective effect on permanent focal cerebral ischemia induced brain injury in rats beyond 6 h treatment time window (Figure 3B-D)” to “These results demonstrated that continuous administration of IMM-H004 (10 mg/kg) showed improved survival compared to Edaravone and Urokinase after permanent focal cerebral ischemia in rats, however the infarct size remains unchanged (Figure 3B-D).”

Comment 2: Line 174-175 - please amend to read "IMM-H004 significantly protected the number of neurons in these brain regions from the stroke-induced reduction (Figure 5B).

Response 2: Thank you very much for your comment.This comment is so prudent. We have amended the substance in line 174-175 “IMM-H004 significantly increased the number of positive neuron cells in these brain regions from the stroke-induced reduction (Figure 5B).” to “IMM-H004 significantly protected the number of neurons in these brain regions from the stroke-induced reduction (Figure 5B).”

Comment 3: Line 264-267 and Figure 11 - define the acronyms and parameters you are measuring for readers not in the field. Mention that this was done at what timepoint post-stroke and post-drug. Mentioned lung infection and inflammation were not found.

Response 3: Thank you very much for your comment.This comment is so important. I have defined the acronyms and parameters, mentioned the timepoint and lung infection were not found in line 267-276 yet.

Comment 4: Related to Comment 4 - modify the manuscript to include your response in the text. I specify this response "needs to be highlighted and discussed".

Response 4: Thank you very much for your comment.This comment is so important. I have modified the manuscript to include my response in the text in line 343-348 yet.

We deeply thank again for the prudent comments on our manuscript.

Very sincerely yours,

Qidi Ai

Round 3

Reviewer 1 Report

The manuscript has been improved.